# Deep Learning- and Expert Knowledge-Based Feature Extraction and Performance Evaluation in Breast Histopathology Images

**DOI:** 10.3390/cancers15123075

**Published:** 2023-06-06

**Authors:** Hepseeba Kode, Buket D. Barkana

**Affiliations:** 1Computer Science and Engineering Department, University of Bridgeport, Bridgeport, CT 06604, USA; hkode@my.bridgeport.edu; 2Electrical Engineering Department, University of Bridgeport, Bridgeport, CT 06604, USA

**Keywords:** CNNs, VGG16, breast cancer, histopathology, knowledge-based, feature extraction

## Abstract

**Simple Summary:**

Breast cancer is one of the leading causes of cancer death among women. Developing machine learning-based diagnosis models receives great attention from researchers and scientists using histopathology images. Deep learning (DL) algorithms automatically extract features from raw data through convolutional operations. The generalization of the DL models’ results relies on large datasets, although they eliminate the expert knowledge in the feature extraction stage. This work aimed to compare the performance of the features extracted via deep learning and a knowledge-based approach in breast cancer detection from histopathology images.

**Abstract:**

Cancer develops when a single or a group of cells grows and spreads uncontrollably. Histopathology images are used in cancer diagnosis since they show tissue and cell structures under a microscope. Knowledge-based and deep learning-based computer-aided detection is an ongoing research field in cancer diagnosis using histopathology images. Feature extraction is vital in both approaches since the feature set is fed to a classifier and determines the performance. This paper evaluates three feature extraction methods and their performance in breast cancer diagnosis. Features are extracted by (1) a Convolutional Neural Network, (2) a transfer learning architecture VGG16, and (3) a knowledge-based system. The feature sets are tested by seven classifiers, including Neural Network (64 units), Random Forest, Multilayer Perceptron, Decision Tree, Support Vector Machines, K-Nearest Neighbors, and Narrow Neural Network (10 units) on the BreakHis 400× image dataset. The CNN achieved up to 85% for the Neural Network and Random Forest, the VGG16 method achieved up to 86% for the Neural Network, and the knowledge-based features achieved up to 98% for Neural Network, Random Forest, Multilayer Perceptron classifiers.

## 1. Introduction

Cancer is one of the most prevalent diseases worldwide. Many reports have shown that breast cancer is the most common cancer among women in the US and worldwide excluding nonmelanoma skin cancer and the second leading cause of cancer-related mortalities after lung cancer. Breast cancer statistics in 2022 reported that the breast cancer rate increased by 0.5% annually but the death rate declined for women except for American Indians and Alaska Natives [1]. The diagnosis can be made by regular mammography or ultrasound checkups. If the imaging indicates the possibility of malignant tissues, a breast tissue biopsy is performed, allowing pathologists to assess the microscopic structure and elements of the tissue histologically. Pathologists perform analysis, which is a challenging task, on nuclei location, mitosis, and glands in classifying benign and malignant tissues [2]. Figure 1 shows normal and ductal carcinoma images from the Human Protein Atlas [3].

At the same time, it is time-consuming, expensive, and dependent on the ability and experience of the pathologists. The appearance of cells in a histopathology image can provide important information for disease diagnosis and prognosis. The characteristics of malignant tissues include inconsistent tissue and cell morphology, uneven color distribution, cell overlapping, and variability in the appearance of stained histological sections, creating difficulties in classification [4]. The characteristics of histopathology images play an essential role in diagnosing cancer. Cell morphology in histopathology images refers to individual cells’ size, shape, and structure of a tissue sample [5]. For example, the size and shape of cancer cells can be different from normal cells. The analysis of cell morphology in histopathology images can also provide information about the activity of specific cellular pathways and processes, such as cell division and apoptosis (programmed cell death). The study of cell morphology in histopathology images is an integral part of precision medicine, where treatment is tailored to the specific characteristics of a disease.

In contrast, uneven color distribution can occur when there are variations in the staining intensity of different regions in the tissue sample. Figure 2 shows various color distributions. For example, in some cancer tissues, certain areas may appear more intensely stained than others, indicating the presence of abnormal cells or tissues. Similarly, cell overlapping in histopathology images refers to the phenomenon where cells in the tissue sample appear on top of one another [6], making it difficult to distinguish individual cells. It can occur when cells have undergone abnormal changes, such as increased cell growth or decreased cell death, leading to tissue overcrowding. Cell overlapping can also be a sign of tissue invasion by cancer cells. All these characteristics of histopathology images can provide valuable information for diagnosis and prognosis.

Histopathology image biomarkers are features extracted from images of tissue samples to diagnose diseases. These features can include morphological changes in cells and tissues and changes in the expression of a specific protein. Preprocessing histopathology images involves a series of steps to improve the quality of raw images to make them suitable for further analysis and interpretation. Typical preprocessing steps are artifact removal, normalization, segmentation, and registration. Normalization ensures that the image data have the same contrast, brightness, and color balance; segmentation separates the ROI from the background, which can be done using techniques such as thresholding, region-based segmentation, or edge detection [5]. Registration aligns the images to a common coordinate system to make them comparable. Image enhancement improves the visual quality of the images using techniques such as contrast stretching, histogram equalization, or denoising. After the preprocessing stage, the next step is extracting the relevant information from the preprocessed images, called features. In machine learning, features can be extracted manually or automatically. Manually extracted features are obtained from Scale-Invariant Feature Transform (SIFT), Speeded Up Robust Features (SURF), Oriented FAST and rotated BRIEF (ORB), image segmentation, and thresholding techniques. Automatic feature extraction can be done by Deep Convolutional Neural Networks (CNN) and pre-trained networks such as ResNet50, VGG16, AlexNet, etc.

## 2. Related Works

Nowadays, artificial intelligence (AI) plays an important role in image classification and designing Computer-Aided Diagnosis (CAD) systems to assist medical professionals in increasing and improving the accuracy of diagnosis and reducing the cost [4,7,8]. CAD systems use machine learning (ML) algorithms to classify benign or malignant images. Image classification is fundamental in computer vision, with numerous medical, biology, and robotics applications. Traditionally, image classification relies on handcrafted features derived from the raw pixel values of the images using domain-specific knowledge and expertise [9,10]. These features capture distinctive characteristics of the region of interest (ROI), such as shape, texture, or color, and are used as input to classifiers such as K-Nearest Neighbors, Support Vector Machines, Decision Tree, etc.

In the last decade, however, deep learning (DL) has emerged as a powerful approach to image classification, providing an alternative to handcrafted features. DL consists of neural networks with multiple layers, which can extract features from the raw pixel values of the images automatically without the need for manual design. Convolutional Neural Networks (CNNs) have been particularly successful and have achieved state-of-the-art performance on many image classification benchmarks [11]. Despite the impressive performance of DL on image classification tasks, it remains debatable which approach is best for a given application, especially in medical fields. On the one hand, handcrafted features can be designed specifically for a given domain and may capture essential characteristics of the ROIs that CNNs do not. On the other hand, CNNs can be more robust and generalizable, as they do not rely on domain-specific knowledge and can learn from large amounts of data.

A recent study defined three sets of features to represent the characteristics of the nuclei cell in breast cancer histopathology images to diagnose malignant cases [9]. The work extracted 33 features based on geometric, intensity, and direction information of nuclei and tested them using the BreakHis histopathology image dataset. Four ML algorithms were designed for diagnosis, Decision Tree, a Narrow Neural Network, a K-Nearest Neighbor, and Support Vector Machines. They acquired the best accuracy for a Narrow Neural Network (NNN) of 96.9%. Another research paper [12] has an improved Autoencoder (AE) network using a Siamese framework designed to learn the compelling features of histopathology images. First, they processed the image features using a Gaussian Pyramid to get the multi-scale, and at the feature extraction stage, a Siamese network was used to constrain the pre-trained AE. The experimental results for their paper showed an accuracy of 97.8% on the BreakHis dataset. Further, researchers have used pre-trained networks such as VGG16, VGG19, AlexNet, and Google Net to classify the histopathology image data. One of the studies proposed a framework called the AlexNet-BC model [13], a pre-trained model using the ImageNet dataset and fine-tuned using an augmented dataset. Further, they used an improved cross-entropy function to make the predictions suitable for uniform distributions, and their approach reached an accuracy of 86.31% on the BreakHis dataset.

The work in [14] demonstrated transfer learning ability for the fully connected network on the histopathology images based on three pre-trained networks of VGG16, VGG19, and ResNet50. Among these, their experimental results gave the highest accuracy of 92.60% for the pre-trained VGG16 network with logistic regression classifier. In contrast, AlexNet and VGG16 models were used in another study [15] that used the BreakHis dataset and fine-tuned the parameters with AlexNet. Then, the obtained features were fed to the SVM classifier. Their evaluation showed that the transfer learning produced better results than the deep feature extraction and SVM classifier. A similar study [16] proposed two different architectures to classify breast cancer from histopathology images. One is single-task CNN architecture to predict malignancy which acquired an average accuracy of 83.25%, and another is a multi-task CNN to predict malignancy and image magnification level, which received an accuracy of 82.13%.

Albashish et al. [17] used VGG16 architecture to extract the features from the breast cancer histopathology images. They used different classifiers for classification purposes and acquired the best accuracy of 96% for the RBF-SVM classifier. Another research paper [18] proposed a dual-stream high-order network named DsHoNewith with six convolutional layers. The first stream uses batch normalization (BN) to keep the original data, and the ghost module is used in another stream to extract high-quality features using linear transformations. Li et al. [19] proposed an embedded fusion mutual learning (EFML) method with two parallel heterogeneous networks for feature extraction and two classifiers, adaptive feature fusion and ensemble classifier, embedded into EFML for classification purposes. Atban et al. [20] used the Resnet18 model for deep feature extraction then they applied Practice Swarm Optimization (PSO), Atom Search Optimization (ASO), and Equilibrium Optimizer (EO) algorithms to histopathology images to obtain more valuable features. For this method, they acquired an accuracy of 97.75% for ResNet18-EO for the SVM classifier with Gaussian and radial-based functions (RBF).

Image classification is a widely studied problem in computer vision, with numerous applications in object recognition, scene understanding, and medical image analysis [21]. One of the key challenges in image classification is how to represent the region of interest in the input images distinctively. It is traditionally done through a process called feature extraction, which involves identifying and extracting relevant features from the images. There are two main approaches to extracting features. (1) a Convolutional Neural Network (CNN) and (2) knowledge-based systems based on domain expertise [9,22].

The CNN-based approach has gained popularity recently due to the success of deep learning models in various tasks [13,23,24]. A CNN is an artificial neural network that processes data with grid-like topologies, such as images [25]. It consists of multiple layers of convolutional and pooling operations, which are used to extract features from the images hierarchically [25]. Later, the extracted features are fed to the classifier for the final prediction. In contrast, knowledge-based feature extraction involves expert knowledge in identifying and extracting distinctive features from the images based on prior knowledge or domain expertise [22]. This process can be time-intensive; sometimes, finding the expertise to design the feature extraction model can be difficult.

One of the main areas of machine learning is called deep learning. The system preprocesses the data to extract features using convolutional layers. Later the network feeds the obtained features to the classifier that predicts whether the given image belongs to the same class. Different techniques are involved in preprocessing the data in each process step. Overfitting data, outliers, dropout, and updating weights and biases are some tasks performed to reduce the loss. Currently, deep learning methods achieved remarkable performance in image processing and computer vision applications. These architectures have automatic feature extraction and representation ability. It has been widely adopted for histopathology image classification tasks [26]. The training process in CNN requires constant parameter adjustments to ensure equivalent learning of all the layers due to overfitting and convergence issues [14]. Some researchers say that transfer learning may be a good substitute for training to overcome the above problems [14]. In transfer learning, the network is trained on different datasets, and its parameters are fine-tuned on the desired dataset. Transfer learning can be used as a baseline or feature generator [27].

Our paper compared the performance of a CNN, transfer learning (VGG16), and knowledge-based feature sets [9] using the BreakHis 400× image dataset. The BreakHis dataset consists of histopathology images of breast tissue samples annotated with ‘benign’ and ‘malignant’ labels. The obtained features were fed to three classifiers: a neural network, Multilayer Perceptron (MLP), and Random Forest (RF). Their performances were compared by using various evaluation metrics.

## 3. Dataset

The BreakHis dataset is a publicly available dataset of histopathological images of breast tissues available at https://www.kaggle.com/datasets/ambarish/breakhis, accessed on 1 September 2022. It has been built in collaboration with the P&D Laboratory-Pathological Anatomy and Cytopathology, Parana, Brazil. It consists of 9109 microscopic images of breast biopsy samples divided into benign and malignant. Each image is labeled with information about tissue type, whether benign or malignant, and the diagnosis given by pathologists. The images are further divided into four magnification levels (40×, 100×, 200×, and 400×) to allow for analysis at different levels of detail. The BreakHis dataset has been widely used for developing and evaluating computer-aided diagnosis (CAD) systems for breast cancer detection. It is considered a benchmark dataset for this task, and several research studies have reported high accuracy rates in detecting breast cancer using the BreakHis dataset. In our research, we are using BreakHis 400× magnification level dataset containing two classes with 547 benign and 1146 malignant images labeled accordingly. Each image is a size of 700 × 460 pixels with 3-channel (RGB) having a depth of 8-bit in each channel and PNG format. BreakHis400× is available at https://www.kaggle.com/datasets/scipygaurav/breakhis-400x-breast-cancer-dataset, accessed on 1 September 2022. This work uses 400× magnification since the knowledge-based method mentioned here [9] extracts features from nuclei such as mitotic count, shape, and texture. Magnifications of 40×, 100×, and 200× do not show these characteristics clearly.

## 4. Methodology

In this paper, we used three approaches to extract the features from the images: (1) a basic CNN architecture, (2) transfer learning, VGG16, and (3) a knowledge-based approach. CNN and transfer learning approaches extract the feature automatically. The origin of the features is not known. Whereas the knowledge-based approach has mathematically defined features, which were driven by expert input. Seven supervised classifiers—Neural Network, Multilayer Perceptron, Random Forest, Decision Tree, SVM, KNN, and NNN- were trained using the feature sets, and their performances were evaluated and compared. Knowledge-based features were obtained from [9], and the details were presented in Section 4.3.

### 4.1. Basic Convolutional Neural Network (CNN) Based Feature Extraction

A Convolutional Neural Network is used for image recognition and preprocessing tasks. Figure 3a,b shows its architecture. It has multiple layers, including convolutional and pooling layers. The idea behind the convolutional layer is to apply a set of filters known as kernels to the input image, resulting in a set of output feature maps. Each filter is applied to a small section of the input image, and the resulting values are summed up to produce a single output value in the feature map. The convolutional layer can detect various features in the input image, such as edges, corners, and blobs using different filters. The max pooling layer is used to reduce the dimensionality of the output feature maps. It works by dividing the feature map into small, non-overlapping regions and taking the maximum value within each region. It helps reduce the feature maps’ spatial size while retaining the most important information.

In our model, we used three convolutional layers along with max pooling layers. The first layer has a set of 32 filters applied to the input image with a kernel size of 3 × 3, activation function ReLU with an input size of 512 × 512, including 3-channel RGB. Then, the max pooling layer is applied with a pooling size of 2 × 2, reducing the spatial size of the output feature maps by taking the maximum value within small, non-overlapping regions. The second layer has the same parameters that the first layers have. The final layer has 64 filters with kernel size 3 × 3, activation function ReLU applied to the output feature maps from the previous layers and produces a final set of output feature maps. A max pooling layer with size 2 × 2 is applied further to reduce the spatial size of the output feature maps.

Furthermore, the dropout rate of 0.25 is applied before flattening to effectively drop out entire channels of feature maps rather than individual neurons. It helps to preserve the spatial information in the feature maps and encourages the network to learn more diverse and robust features. After flattening, the feature vector will be fed to classifiers, as shown in Figure 3a.

### 4.2. Transfer Learning (VGG16) Based Feature Extraction

The VGG16 [25] model consists of 16 layers of convolutional and pooling operations and has been widely used for image classification tasks. It is characterized by using small 3 × 3 filters throughout the network, which leads to a deeper network with smaller receptive fields. We extracted features from the images using the VGG16 CNN model, which was trained on the ImageNet dataset [28]. First, the pre-trained VGG16 model was loaded and did not include the top layers, as these were the layers responsible for the final classification. Then, a new model was created, which took the output of the last layers of VGG16 as input. Next, the preprocessed BreakHis image dataset was fed to the model to extract the features from the images. These extracted features were then fed to the classifiers. Figure 4 illustrates the transfer learning-based feature extraction steps.

The below shows the implementation steps of the VGG16 model used in this study.
Steps: Feature Extraction from Input Images Using the VGG16 Model.Input:BreakHis 400× training image dataPreprocessing:Resizing the images to 512 × 512Normalizing pixel values to between 0 and 1Output:Initialize the VGG16 model.Import VGG16 model
Remove the fully connected layers.Load the VGG16 model for feature extraction by removing the FC layers as they were designed for ImageNet classification tasks.VGG_model = VGG16 (weights = ’Imagenet’, include_top = False, input_shape = (512, 512, 3)3-is the RGB channel since we are using color imagesPass the input images through the model:Forward pass the preprocessed images through the VGG16 model to obtain the output feature maps.Retrieve the output of the last convolutional layer as they capture the high-level abstract features.Flatten the features.Flatten(img): Convert the 3-D extracted features to the 1-D feature vectorStore the extracted features with corresponding image labels for classification tasksFeatures(img) = Flatten(img)

### 4.3. Knowledge-Based Feature Extraction

For this approach, the features were calculated based on domain expertise. It involved analyzing the images and identifying specific characteristics or patterns indicative of the class label. In this case, the features include the unique characteristics of cell nuclei representing malignant cases. We adopted this knowledge-based feature set from [9]. Figure 5 illustrates the overall structure of the knowledge-based system.

Three sets of features were calculated: Geometric, Directional, and Intensity which include 33 features together to define the characteristics of the cell nuclei for each image. Here, we summarize these feature sets.

#### 4.3.1. Geometrical Features

In [9], the geometric features have been calculated by the parameters such as area (A), perimeter (P), roundness (R), and the number of segmented cells (*L*). *A* is the area of the nucleus, where $x_{k}$ is the number of segmented pixels of each nucleus, *P* is the perimeter of the nucleus where $P_{k}$ is the number of pixels at the cell border of each nucleus, $\frac{A}{P}$ is the ratio of the area to the perimeter, and *R* is the roundness of the ROI calculated from the area and perimeter.
(1)$$f_{1}:L$$
(2)$$f_{2}:A=\frac{1}{L}\sum_{k=1}^{L}x_{k}\;for\;k=1,2,\ldots ,L$$
(3)$$f_{3}:P=\frac{1}{L}\sum_{k=1}^{L}P_{k}$$
(4)$$f_{4}:\frac{A}{P}$$
(5)$$f_{5}:R=\frac{4A\pi }{{\left(P\right)}^{2}}$$

#### 4.3.2. Directional Features

It is known that malignant cells have irregular shapes, whereas benign cells have more definite and round forms. Spatial distance in every eight directions, East (*E*), West (*W*), North (*N*), South (*S*), Northwest (*NW*), Northeast (*NE*), Southwest (*SW*), and Southeast (*SE*), are calculated to determine the shape and state of the segmented nuclei.
(6)$$d_{N}=\sqrt{{\left(m-m^{\prime }\right)}^{2}+{\left(n-n^{\prime }\right)}^{2}}$$
(7)$$f_{6}\to f_{13}:{\overset{-}{m}}_{D}=\frac{1}{L}\sum_{k=1}^{L}d_{D_{k}}$$
D=N,S,E,W,NW,NE,SW,SE

${\overset{-}{m}}_{D}$ is mean for a particular direction of the segmented region. The standard deviation represents the variation in the eight directions (*D*). $\sigma_{D}$ is the standard deviation for each cardinal direction.




$f_{14}\to f_{21}:$



(8)
$$\sigma_{D}=\sqrt{\frac{\sum_{k=1}^{L}{\left(d_{D_{k}}-{\overset{-}{m}}_{D}\right)}^{2}}{L}}\;for\;D=N,S,E,W,NW,NE,SW,SE$$



The range is calculated as the difference between the eight directions’ maximum and minimum spatial distance values. This feature set will make up of features $f_{22}\to f_{29}$.




$f_{22}\to f_{29}:$



$$range\left(D_{k}\right)=\max\left(d_{D_{k}}\right)-\min\left(d_{D_{k}}\right)$$


(9)
for k=1,2,…,L and D=N,S,E,W,NW,NE,SW,SE



The ratio of the segmented region to the area of the fitting rectangle $\left(A_{R}\right)$ is defined to represent the directional feature set.
(10)$$f_{30}:AR_{ratio}=\frac{1}{L}\sum_{k=1}^{L}\frac{A_{k}}{A_{R_{k}}}$$

#### 4.3.3. Intensity-Based Features

Intensity-based features capture mitotic cell-related details. The intensity-based features include mean, standard deviation, and range measurements.
(11)$$f_{31}:\overset{-}{I}=\frac{1}{L}\sum_{k=1}^{L}I_{k}$$

Here, I is the average intensity of a nucleus.
(12)$$f_{32}:\sigma_{I}=\frac{1}{L}\sum_{k=1}^{L}{{\sigma_{I}}_{k}}_{}$$
(13)$$\sigma_{I_{k}}=\sqrt{\frac{\sum_{i=1}^{x_{k}}{\left(J_{k}\left(i\right)-I_{k}\right)}^{2}}{x_{k}}}$$

$R_{I}$ is the intensity range that is calculated by the average value of the difference between the maximum and minimum pixel intensity values of nuclei.
(14)$$f_{33}:R_{I}=\frac{1}{L}\sum_{k=1}^{L}{R_{I}}_{k}$$
(15)$$R_{I_{k}}=\max\left(J_{k}\right)-\min\left(J_{k}\right)\;\;for\;k=1,2,\ldots L$$

The calculated features were analyzed in terms of their efficiency in separating benign and malignant cases. We ranked the 33 features using the minimum redundancy maximum relevance (MRMR) algorithm. Figure 6 illustrates the feature ranking. The drop in the importance score represents the confidence in the selection algorithm. There is a large drop between the first two and the remaining features. The mean intensity value of the intensity-based set was found to be the second most significant feature. The ranking order was as follows: Feature number: 27, 31, 18, 24, 30, 29, 8, 21, 23, 15, 26, 25, 28, 12, 20, 3, 5, 6, 22, 11, 9, 17, 16, 14, 4, 19, 10, 13, 7, 32, 1, 2, 33. However, overall, directional features were found to be more significant than the geometrical and intensity-based features as 14 out of the 15 features in the ranking belonged to the directional features that were developed to represent the irregular border shape of the segmented cells in an image. Details of the features can be found in [9]. Figure 7 and Figure 8 plot some of the features versus each other to show the benign and malignant cases. As seen in the figures, the represented directional and intensity features distinguish two classes with some overlapping. All features were fed to the classier in this work.

## 5. Classification and Performance Evaluation

We evaluated the performance of the feature sets using seven different classifiers: a Neural Network (NN) (64 units), a Random Forest (RF), a Multilayer Perceptron (MLP), Decision Tree, Support Vector Machine (SVM), KNN, Narrow NN (10 units).

### 5.1. Neural Network (NN)

The Neural Network (NN) classifier is widely used for image classification tasks. It consists of multiple layers of interconnected “neurons” that are used to process and analyze the input data. The NN is trained using a large dataset of labeled images and adjusts the weights and biases of the neurons to optimize the prediction performance. In our experiment, we designed an NN where a dense layer with 64 units and a ‘ReLu’ activation function is added, followed by a final dense layer with one unit and a ‘Sigmoid’ activation function. The model was compiled with the ‘Adam’ optimizer and binary cross entropy loss function and trained with extracted features and corresponding labels.

### 5.2. Multilayer Perceptron (MLP)

This type of neural network classifier consists of multiple layers of neurons with a feedforward architecture. It is used for image classification tasks due to its ability to learn complex patterns in datasets. The MLP classifier is trained by adjusting the weights and biases of the neurons through an optimization process, such as backpropagation. In our study, we designed MLP classifier with ‘hidden_layer_sizes = 100’, ‘max_iter = 500’, and a ‘ReLu’ activation function.

### 5.3. Random Forest (RF)

Random Forest classifiers use an ensemble learning method that combines multiple Decision Trees and use the average prediction of all the trees to make a final prediction. It is widely used for image classification tasks due to its ability to handle high-dimensional data. A bootstrap sample is taken from the original data set with a replacement for each tree in the forest. It means that some data points may be included multiple times in a given bootstrap sample. From this sample, a Decision Tree is constructed using a random subset of features at each node. It helps to reduce the correlation between trees and improves the overall performance of the forest. Once all trees have been constructed, the Random Forest predicts a given input by averaging the predictions of all the trees. In our work, we imported a Random Forest classifier from the scikit-learn library and used the parameter ’n_estimators = 100’ for all three feature extraction approaches.

### 5.4. Decision Tree

The Decision Tree is one of the popular supervised classifiers. It can be defined by decision nodes and leaves, where the leaves are the decisions. In this work, the fine tree classifier was set up with the same parameters used in [9]. The maximum number of splits: 100. Split criterion: Gini’s diversity index and Surrogate decision splits: off.

### 5.5. Support Vector Machines (SVM)

We set the SVM parameters as follows. The kernel function was set as linear, and the kernel scale was set to automatic. The box constraint level was set to 1, and the multiclass method was set to One-v-One. The value for the standardized data was set to true.

### 5.6. KNN

The Fine KNN classifier was used in our work as the number of neighbors was set to 1, and the distance metric was set to Euclidean. The distance weight and the standardized data values were set to Equal and true.

### 5.7. Narrow Neural Networks (NNN)

The Narrow Neural Network had one fully connected layer with a size of 10. The activation used was ReLU with an iteration limit of 1000. The regularization strength (Lambda) was set to 0.

### 5.8. BreakHist Dataset and Evaluation Metrics

We used the BreakHis 400× Histopathology image dataset, which consists of 547 benign and 1146 malignant images Spanhol et al. 2016 [29]. For this dataset, the images were resized to (512 × 512) and split the data into a training set and testing set of 80% and 20%, respectively. Performance evaluations were done by accuracy, precision, recall, F1 score, and ROC curves. All these metrics are based on confusion matrices. The metric parameters are computed using true negative (*TN*), true positive (*TP*), false positive (*FP*), and false negative (*FN*). *TP* refers to the cases where the model predicted the positive class and the actual class is positive; similarly, *TN* refers to the cases where the model predicted the negative class and the actual class is also negative. Whereas *FP* refers to the cases where the model predicted the positive class, the actual class is negative. It is known as the Type 1 error. Similarly, *FN* refers to the cases where the model predicted the negative class, but the actual class is positive. It is known as the Type 2 error. A confusion matrix evaluates the performance of a classification model by providing a detailed picture of how well the model is performing. In medical diagnosis, minimizing *FN* is often more important. The accuracy is calculated as
(16)$$\mathrm{Accuracy}=\frac{TN+TP}{TN+TP+FN+FP}$$

Precision and Recall are other widely used metrics to evaluate the performance of the testing data. Precision and Recall can be calculated as follows:(17)$$Precision=\frac{TP}{TP+FP}$$
(18)$$\mathrm{Recall}=\frac{TP}{TP+FN}$$

The other evaluation metric is F1-score, which is commonly used for statistical measures. The values of this F1-score range from 0 to 1. It can be calculated as follows:(19)$$F1\text{-}\mathrm{score}=2*\frac{\mathrm{Precision}*\mathrm{Recall}}{\mathrm{Precision}+\mathrm{Recall}}$$

## 6. Experimental Results

Table 1 presents the performance of the proposed methodology. The presented values represent the macro average with ±0.02 tolerance, which supervised classifiers’ performances slightly change depending on the training data. As can be seen, the performance of the knowledge-based feature extractor consistently outperformed the CNN and VGG16-based feature extractors using three classifiers on the BreaKHis 400× dataset.

We employed the BreakHis 400× dataset for a couple of reasons. (1) Our work compares the knowledge-based features with CNN and VGG16-based features. Knowledge-based features are defined for histopathology images with 400× magnification since it shows the nuclei characteristics and segmentation of nuclei, which is still challenging due to overlapping and clustered nuclei. (2) The dataset provides binary classification, and our work proposed three models for binary classification.

The ROC (Receiving Operating Characteristic) curve is a graphical representation of the performance of a binary classifier at a different classification threshold. It is created by plotting the *TP* rate against the *FP* rate for various threshold values. The TPR is the proportion of actual positives correctly identified as such, while the FPR is the proportion of actual negatives incorrectly identified as positives. The ROC curve is a useful tool for evaluating and comparing the performance of binary classifiers. The closer the ROC curve is to the top left corner, the better the performance of the classifier. AUC (Area Under the Curve) is a metric that measures the overall performance of a binary classifier. It represents the area under the ROC curve and is interpreted as the probability that a randomly chosen positive example will be ranked higher than a randomly chosen negative example. The AUC value ranges from 0 to 1, with a value of 0.5 indicating a random classifier and a value of 1 indicating a perfect classifier. The higher AUC value indicates a better overall performance of the classifier. The ROC and AUC curves provide a graphical and numerical measure of the classifier’s performance, making them helpful in evaluating and comparing different classification models. Figure 9, Figure 10 and Figure 11 show the ROC and AUC of the CNN, VGG16, and knowledge-based feature extraction approaches. We present the convergence performance of the employed models in Figure 12.

Table 2 compares the performance of the proposed method with the previously proposed methods in literature on the BreaKHis 400× dataset. The results show that the evaluated knowledge base feature set outperformed the feature sets extracted by the ML-based features.

The results of our evaluation demonstrate the high capability of the knowledge-based approach to feature extraction for image classification tasks. It is time intensive, relies on expert knowledge for feature definition and selection design, and extracts high-quality features. In contrast, automatically extracted features by CNN and VGG16 were less effective than the knowledge-based features.

One possible reason for the superior performance of the knowledge-based approach is identifying each characteristic of the cell nuclei based on expert opinion and designing the model to extract that information from the cell nuclei, which determines whether the cell is malignant. Another reason is robustness to noise and variability. Histopathology images can be subject to noise and variability, such as differences in staining techniques or tissue preparation. Knowledge-based approaches can help identify features robust to these variations, leading to more accurate and consistent results. These features are often based on biological structures and processes, which can provide greater interpretability and insight into the underlying biology of the tissue. It is particularly important since understanding the biological basis of the disease is crucial for developing effective treatments.

The size of the dataset does not affect the knowledge-based feature calculations. The features’ mathematical model was developed based on content knowledge. The classifier’s performance may change because of the type of classifier design and specifications, such as Support Vector Machines, KNN, ANN, CNN, etc. However, we can employ the knowledge-based features to design a rule-based classifier such as a Fuzzy classifier. In contrast, dataset size affects the features calculated via CNN and VGG16. It is the major disadvantage of these two models. To highlight the limitation of such models, our study compares the capabilities of the three feature sets coming from traditional CNN, transfer learning, and knowledge-based.

The CNN models performed well on large datasets of images and learned features well-suited for image classification tasks, as shown in previous studies [25]. Some research reported that CNN-based and VGG16-based feature extraction methods were more effective than knowledge-based feature extraction. It could be because their handcrafted features were not as unique and informative as those learned by CNN and VGG16 models [38]. Here, we used three knowledge-based feature sets that were initially developed by Labrada in 2022. The 33 features represent the cell nuclei’s morphological, directional, and orientation [9]. The VGG16 feature extraction method also performed better when used with a simple Neural Network than MLP and RF.

In contrast, it performed poorly compared with knowledge-based feature extraction. It could be because the VGG16 model was not explicitly developed for histopathology image classification tasks. Our results are consistent with previous studies showing that fine-tuning a pre-trained CNN model can improve performance on image classification tasks [39].

The works in [18,19] employed DsHoNet and EFML methods, which achieved high classification performance. The research approach of using a dual-stream high-order network (DsHoNet) for breast cancer classification can be considered efficient in terms of its potential to improve accuracy and performance. By employing a dual-stream architecture, the model can leverage multiple sources of information or features extracted from histopathology images, enhancing the classification process. Additionally, incorporating high-order network components allows for capturing complex relationships and patterns within the data, potentially leading to improved classification results. However, the actual efficiency of this research would depend on factors such as the size and diversity of the dataset, the training methodology, and the evaluation metrics used to assess the performance of the classification model.

However, DsHoNet can generally be considered a relatively complex model compared to more straightforward approaches. The use of a dual-stream architecture implies the presence of two separate pathways or streams within the network, each handling different types of input or extracting different types of features. It introduces additional complexity compared to single-stream models.

Furthermore, incorporating high-order network components suggests the model captures higher-order interactions or dependencies among features. It typically involves modeling complex relationships and may require more computational resources and advanced techniques.

Overall, DsHoNet is likely to have a higher complexity level than traditional models, as it aims to leverage multiple streams of information and capture more intricate relationships within the data.

EFML involves designing a network architecture that integrates and learns from multiple modalities or feature sources. It typically requires the development of specialized network modules or fusion strategies to combine information from different sources. The complexity of the architecture depends on factors such as the number and type of modalities, the fusion approach used (early fusion, late fusion, or intermediate fusion), and the interactions between modalities within the network.

EFML involves training a model with multiple modalities or feature sources, which increases the computational requirements compared to single-modal approaches. The training process may include backpropagation and optimization across multiple branches or streams, potentially leading to increased training time and memory usage. Additionally, the computational complexity can increase if the modalities have different resolutions or require separate preprocessing steps. Our work’s models have less complexity than DsHoNet and EFML.

Work in [11] used ResNet50, achieved an accuracy of 93%, and did not report other performance metrics. [20] used ResNet18 and achieved accuracies of 90%, 95%, and 75 using KNN, SVM, and Decision Tree classifiers, respectively. ResNet50 is undoubtedly an advanced model with remarkable results in many applications. Although many studies proved that the Resnet50 model performs better than VGG16, detecting errors becomes difficult due to its deeper network. Furthermore, if the network is shallow, the learning might be inefficient. Medical image datasets need to build efficient learning models to predict the classes. In our research, we wanted to use VGG16 due to its very small receptive fields instead of massive fields, such as Resnet50, for the BreakHis dataset. VGG16 has its own merits, making it a suitable choice for our research. VGG16 is a deep Convolutional Neural Network, and its simplicity and straightforward design make it easier to understand and implement, which was beneficial for our research objectives. Additionally, VGG16 has demonstrated strong performance across various image classification benchmarks, and its pre-trained weights are readily available, allowing for efficient transfer learning.

Several directions for future work would build on the results of this work. One possible way is to explore using different transfer learning and CNN models for feature extraction. Many CNN models are available, each with unique and various characteristics and capabilities. It would be an exciting topic to explore and compare the performance of these models on different image classification methods to determine which is most effective. Another work can examine the impact of other preprocessing techniques on the performance of the CNN- based approach. The preprocessing stage impacts the extracted features’ quality and the classifier’s overall performance. Therefore, it would be helpful to study the effect of different preprocessing techniques on the performance of the CNN-based approach. Finally, it would be interesting to study the effectiveness of CNN-based and knowledge-based feature extraction and their variations in the training data. It would be helpful to evaluate the performance of the approaches when the data are limited or biased in some way. It could provide insights into the generalization capabilities of the CNN- and knowledge-based systems.

We are implementing a Vision transformer (VIT) for breast cancer diagnosis from the histopathology images as an ongoing work.

Among different image-based datasets, histopathology images are one where deep neural network techniques perform well in diagnosing benign and malignant tissues. However, the present mainstream techniques face challenges. One of them is that the high-resolution characteristics of histopathological images have not been fully utilized to improve classification accuracy. One reason is that the current patch-based method does not adequately integrate these patches to make the classification result from a complete histopathology image [4].

## 7. Conclusions

In this study, we designed CNNs, VGG16, and knowledge-based models for feature extraction. We evaluated their performance on the BreakHis 400× dataset using Neural Networks (64 units), Multilayer Perceptron, Random Forest, Decision Tree, Support Vector Machines, K-Nearest Neighbor, and Narrow Neural Network (10 units) classifiers. The basic CNN-based features achieved an accuracy of up to 85%. The VGG16-based features attained an accuracy of up to 86%. Knowledge-based features outperformed the CNN and VGG16 features significantly. It reached an accuracy of 98% for all classifiers mentioned above. Without depending on the size of the dataset and training, the features were calculated by mathematical equations that were defined to capture expert knowledge. Overall, our work reports that the knowledge-based CAD system produced a robust high performance in benign/malignant classification using histopathology images.

## Figures and Tables

**Figure 1 cancers-15-03075-f001:**
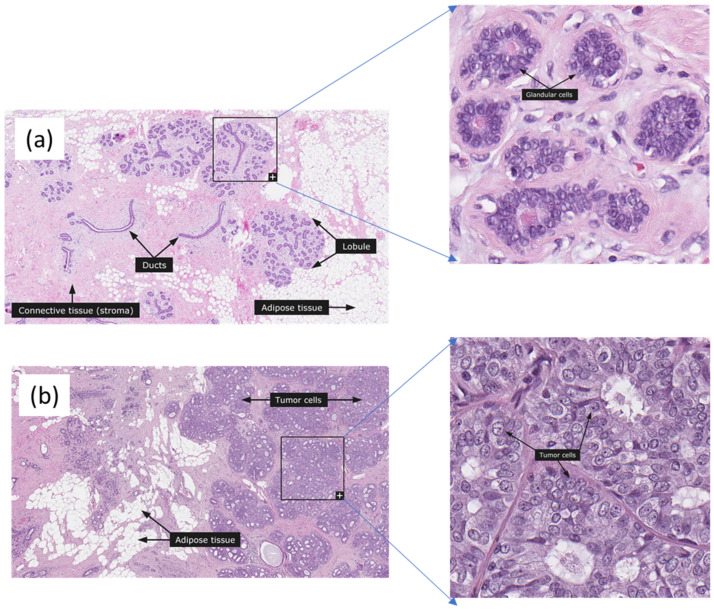
Normal and ductal carcinoma breast histopathology images from the Huma Protein Atlas Tumor cells, adipose tissue, and glandular cells are marked. (**a**) Normal, (**b**) Ductal Carcinoma, 68 years old female, Elston-Ellis score is 4. The images were used with the permission of The Human Protein Atlas; this work met their conditions for the use of the above images.

**Figure 2 cancers-15-03075-f002:**
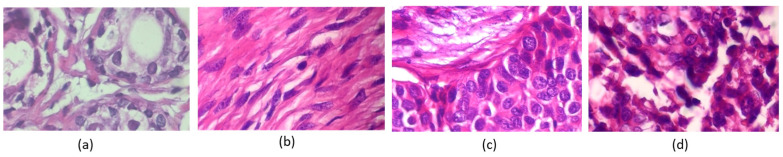
Breast cancer histopathology images from the BreakHis 400× image dataset. The images in (**a**,**b**) are classified as benign, and the images shown in (**c**,**d**) are classified as malignant by pathologists.

**Figure 3 cancers-15-03075-f003:**
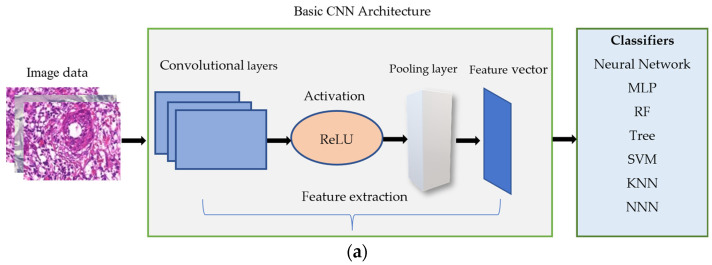
(**a**) The architecture of extracting the features by basic CNN model and feeding them to the three classifiers. (**b**) The details of the architecture.

**Figure 4 cancers-15-03075-f004:**
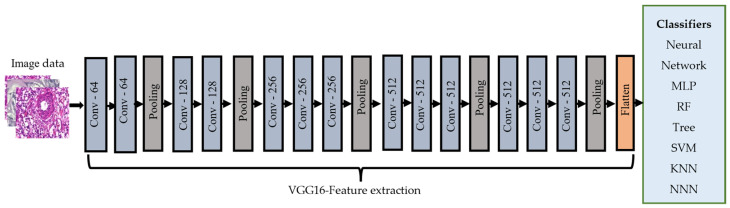
The VGG-16 model was used in our approach.

**Figure 5 cancers-15-03075-f005:**
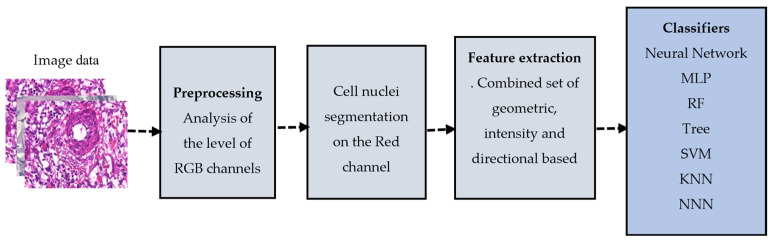
Knowledge-based computer-aided model.

**Figure 6 cancers-15-03075-f006:**
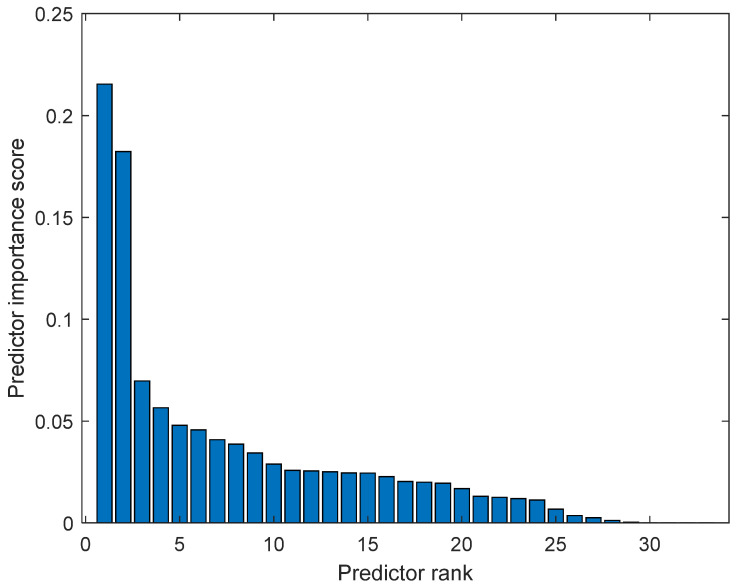
Feature ranking using minimum redundancy maximum relevance (MRMR) algorithm.

**Figure 7 cancers-15-03075-f007:**
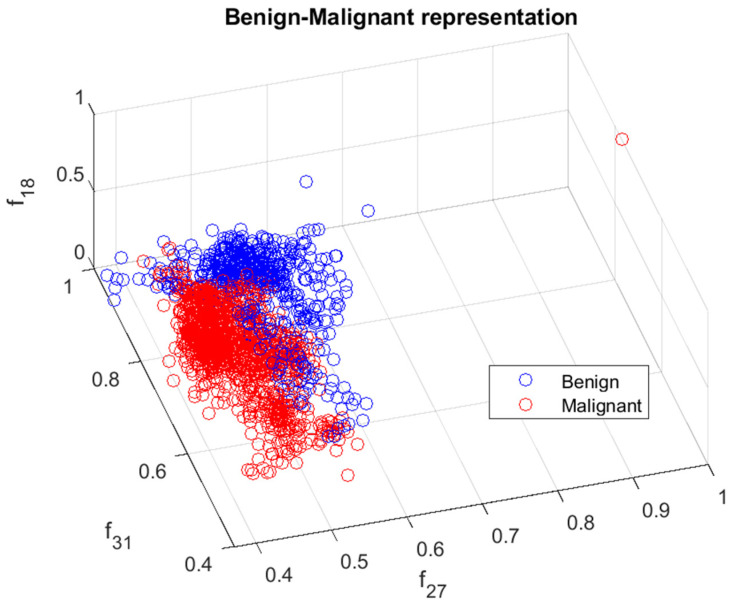
Benign and malignant characterization by directional features $f_{18},f_{27}$ and intensity-based feature $f_{31}$. Benign and malignant cases are represented in blue and red, respectively.

**Figure 8 cancers-15-03075-f008:**
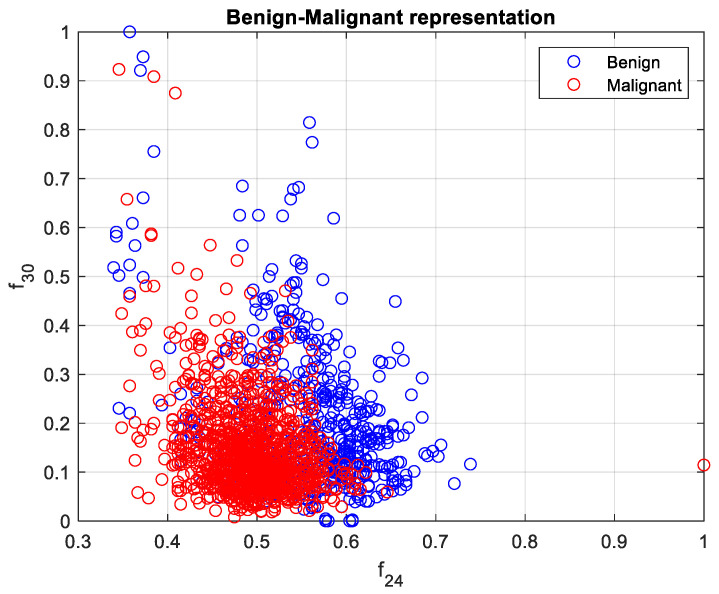
Benign and malignant characterization by two directional features $f_{24}\;and\;f_{30}$. Benign and malignant cases are represented in blue and red, respectively.

**Figure 9 cancers-15-03075-f009:**
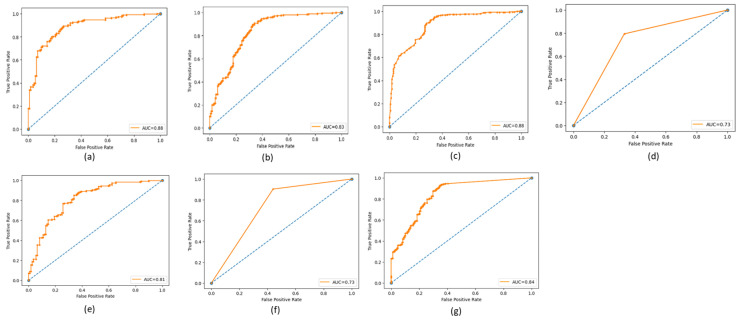
ROC and AUC graphical representations of the CNN feature extraction approach using the classifiers. (**a**) NN (64 units), (**b**) MLP, (**c**) RF, (**d**) Decision tree, (**e**) SVM, (**f**) KNN, and (**g**) NNN (10 units).

**Figure 10 cancers-15-03075-f010:**
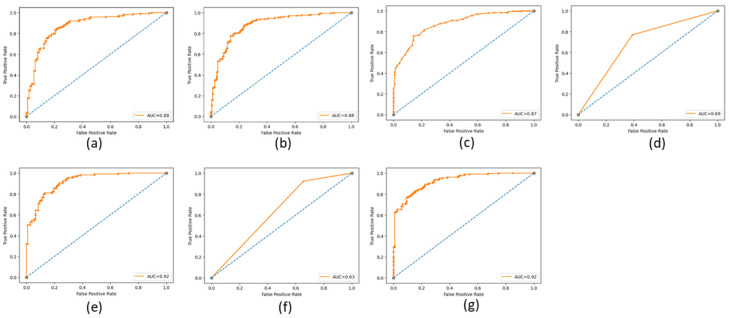
ROC and AUC graphical representation of VGG16 feature extraction approach using the classifiers. (**a**) NN (64 units), (**b**) MLP, (**c**) RF, (**d**) Decision tree, (**e**) SVM, (**f**) KNN, and (**g**) NNN (10 units).

**Figure 11 cancers-15-03075-f011:**
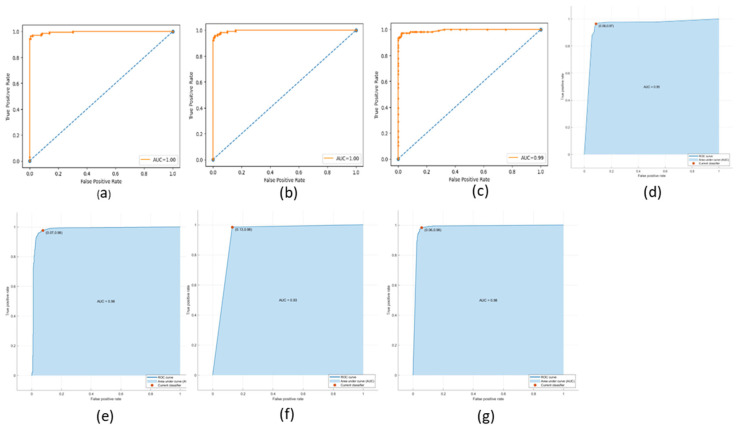
ROC and AUC graphical representation of knowledge-based feature extraction approach using the classifiers. (**a**) NN (64 units), (**b**) MLP, (**c**) RF, (**d**) Decision tree, (**e**) SVM, (**f**) KNN, and (**g**) NNN (10 units).

**Figure 12 cancers-15-03075-f012:**
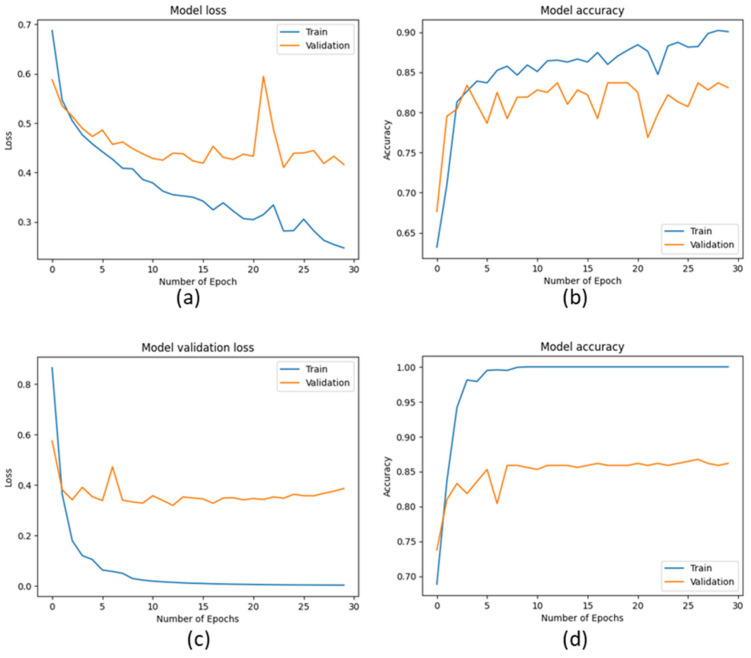
Model loss and accuracy performance for training and validation data. (**a**,**b**) CNN (**c**,**d**) VGG16.

**Table 1 cancers-15-03075-t001:** Macro average performances of the proposed approaches (±0.02)—VGG16, CNN, and knowledge-based—for binary classification using Neural Networks (64 units), Multilayer Perceptron, Random Forest, Decision Tree, Support Vector Machines, K-Nearest Neighbor, and Narrow Neural Network (10 units) classifiers on the BreakHis 400× dataset.

Feature Extractor	Classifier	Accuracy	Precision	Recall	F1 Score
Basic CNN	Neural Network (64 units)	0.85	0.84	0.82	0.83
	MLP	0.84	0.84	0.79	0.81
	RF	0.85	0.85	0.79	0.81
	Tree	0.75	0.72	0.73	0.73
	SVM	0.79	0.76	0.75	0.76
	KNN	0.79	0.77	0.73	0.75
	NNN (10 units)	0.84	0.83	0.79	0.8
VGG16	Neural Network (64 units)	0.86	0.87	0.81	0.83
	MLP	0.82	0.83	0.76	0.88
	RF	0.74	0.84	0.62	0.61
	Tree	0.73	0.7	0.7	0.7
	SVM	0.87	0.86	0.83	0.84
	KNN	0.73	0.71	0.63	0.64
	NNN (10 units)	0.8	0.86	0.67	0.69
Knowledge-based extraction	Neural Network (64 units)	0.98	0.98	0.96	0.97
	MLP	0.98	0.98	0.97	0.98
	RF	0.98	0.98	0.96	0.98
	Tree [9]	0.85	0.9	0.88	0.89
	SVM [9]	0.96	0.9	0.88	0.89
	KNN [9]	0.94	0.98	0.94	0.96
	NNN [9]	0.97	0.98	0.98	0.98

**Table 2 cancers-15-03075-t002:** Comparison of the state-of-art works on 400× BreakHis dataset with binary class classification.

Method	Feature Extraction	Classifier	Accuracy	Precision	Recall	F1 Score
Labrada et al., 2022 [9]	Knowledge-based [9]	Decision Tree	0.85	0.9	0.88	0.89
		SVM	0.96	0.9	0.88	0.89
		KNN	0.94	0.98	0.94	0.96
		NNN	0.97	0.98	0.97	0.98
Sharma and Mehra 2020 [10]	VGG16	RF	0.69	0.7	0.69	0.69
		SVM	0.92	0.92	0.91	0.91
		LR	0.85	0.86	0.86	0.86
Sharma and Mehra, 2020 [10]	Hu moment, Colored histogram, and Haralick texture	RF	0.86	-	-	-
		SVM	0.83			
Gupta et al., 2020 [11]	ResNet50	Linear Regression	0.93	-	-	-
		SVM	0.93			
Deniz et al., 2018 [15]	VGG16+Alexnet	Fully connected layers (FC6)	0.87	-	-	-
Albashish et al., 2021 [17]	VGG16	RBF-SVM	0.96	-	-	-
		NN	0.9			
Zou et al., 2022 [18]	DsHoNet	-	0.98	0.98	0.98	0.98
Li et al., 2023 [19]	EFML	-	0.99	0.99	0.99	0.99
Atban et al., 2023 [20]	ResNet18	KNN	0.9	0.9	0.9	0.9
		SVM	0.95	0.95	0.95	0.95
		Decision Tree	0.75	0.75	0.77	0.76
Spanhol et al., 2016 [29]	PFTAS	SVM	0.82	-	-	-
		RF	0.81			
Zhang et al., 2018 [30]	CNN	SVM	0.78	-	-	-
		RF	0.75			
		KNN	0.75			
Spanhol et al., 2017 [31]	CaffeNet	Neural network (Fully connected layers (fc7))	0.82	-	-	0.87
Fabio Alexandre Spanhol et al., 2016 [32]	CNN	Neural network (Fully connected layers)	0.8	-	-	-
Adeshina et al. nd. [33]	Deep CNN	Neural network (Fully connected layers	0.91	0.63	0.77	-
Gour et al., 2020 [34]	ResHist features	RF	0.86	-	-	-
		SVM	0.86			
Gour et al., 2020 [34]	VGG16	Simple Neural network (with fully connected layers)	0.81	-	-	-
Togacar et al., 2020 [35]	BreastNet	ML classifier	0.96	0.95	0.95	0.95
Zhu et al., 2019 [36]	Hybrid CNN model	-	0.81	0.82	0.93	0.87
Zou et al., 2021 [37]	AHoNet	-	0.96	0.95	0.96	0.95
**This work**	**CNN**	**Neural network with FC layers**	**0.85**	**0.84**	**0.82**	**0.83**
	**VGG16**		**0.86**	**0.87**	**0.81**	**0.83**
**This work**	**Knowledge-based**	**Neural Network**	**0.98**	**0.98**	**0.96**	**0.97**
		**MLP**	**0.98**	**0.98**	**0.97**	**0.98**
		**Random forest**	**0.98**	**0.98**	**0.96**	**0.98**

## Data Availability

Our work does not have a publicly archived dataset.

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
