# Peer review of "Deep Learning- and Expert Knowledge-Based Feature Extraction and Performance Evaluation in Breast Histopathology Images"

_cancers, 2023, doi:10.3390/cancers15123075_

Round 1
Reviewer 1 Report
Some recommendations to the authors:
Page 2, Figure 1: From where were the pictures taken? As a pathologist, I don’t recognize malignancy in pictures “c” or “d”
Reference 4 should not be used to support the statements in the “Introduction”. In fact, the first part of the introduction related to epidemiology and pathology of breast cancer should not have references from papers related with computer vision.
Figure 2: Needs an expert for annotations. For instance, the last annotation form “e” is not adipose tissue but a blood vessel. In “f”, the “mitotic figure” is absent and the “adipose tissue” is just a retraction artefact.
Where is Figure 8? I couln´t find it on the manuscript.
The work needs a better description of the BreakHis datset to be fully understood.
Additionally, the authors could test the model in a different dataset such as the BACH dataset.
The work with reference “9” (which I think this work is an improvment of) could also be included in this paper to enrich and to improve understandability.
The works with reference “18” and “19” achieve an equal and higher accuracy compared to the current study and there is no reference to this in the discussion.
This work should include an attempt to explain how this models work.
Additionally, it would be extremally important to the field of cancer science, if the authors could inform about the morphologic characteristics most important to the classification of benign and malignant cases with their models.
Minor editing of English language required
Author Response
We thank you for the opportunity to revise “Deep Learning- and Expert Knowledge-based Feature Extraction and Performance Evaluation in Breast Histopathology Images”. We are pleased to submit the revised manuscript. We appreciated the constructive criticisms of the reviewers. This document outlines how we have addressed them. All the revisions are written in blue in the manuscript.
---------------------------------------------------------------------------------------------------------------------------------------
Reviewer 1:
Q1. Page 2, Figure 1: From where were the pictures taken? As a pathologist, I don't recognize malignancy in pictures "c" or "d"
Thank you. We reorganized the figures. The questioned figure is Figure 2 now. The images are from the BreakHis dataset. The benign and malignant cases are labeled by pathologists and provided as the ground truth by the dataset. We added a paragraph to clarify it.
Q2. Reference 4 should not be used to support the statements in the "Introduction". In fact, the first part of the introduction related to the epidemiology and pathology of breast cancer should not have references from papers related to computer vision.
Thank you. We changed the references in the introduction, and first part and used proper sources. We kept reference 4 and cited it on Page 2; the presented information was related to classification.
Q3. Figure 2: Needs an expert for annotations. For instance, the last annotation form "e" is not adipose tissue but a blood vessel. In "f", the "mitotic figure" is absent and the "adipose tissue" is just a retraction artifact.
Thank you. We replaced Figure 2, which shows experts' annotations. It is now Figure 1 in the revised version.
Q4. Where is Figure 8? I couldn't find it in the manuscript.
Thank you. We fixed the issue.
Q5. The work needs a better description of the BreakHis dataset to be fully understood.
Thank you for your suggestion. We added a section (Section 3) to describe the BreakHis dataset on page 5. We adjusted the following section numbers accordingly.
Q6. Additionally, the authors could test the model in a different dataset, such as the BACH dataset.
Thank you. The BACH dataset has four classes, including normal, benign, in-situ, and invasive. It is a small dataset with 430 images with no defined magnification value. Our research work was designed for a binary classification as benign or malignant. To be able to use the BACH dataset, the proposed CAD architecture needs to be redesigned. Even if it is done, the discussion section will be challenging to compare binary and multi-class results. Most of the research uses the BreakHis 400x dataset since it shows the nuclei characteristics and segmentation of nuclei from 400x is still challenging due to overlapping and clustered nuclei.
We would like to keep our work's focus on binary classification.
Q7. The work with reference "9" (which I think this work is an improvement of) could also be included in this paper to enrich and improve understandability.
Thank you. Our work adapts the features developed by Reference 9. We elaborated on the calculated features and their significance in separating benign and malignant cases. A feature ranking section is added on pages 10 and 11. As an example, we plotted several features versus each other to show the classes.
Q8. The works with reference "18" and "19" achieve an equal and higher accuracy compared to the current study and there is no reference to this in the discussion.
Thank you. We added a discussion in Section 6.
Q9. This work should include an attempt to explain how this models work. Additionally, it would be extremally important to the field of cancer science, if the authors could inform about the morphologic characteristics most important to the classification of benign and malignant cases with their models.
Thank you. Deep neural networks and transfer learning methods use convolutional layers to extract features automatically. The origin of the features is not known, as if the features represent texture or morphological characteristics. However, we can analyze the impact of the knowledge-based features. Directional features capture morphological and homogeneous characteristics of each nucleus in eight directions. We expanded Section 4C and reported that the directional features had the most impact on benign/malignant classification.
Reviewer 2 Report
In this manuscript, the authors designed three CAD systems and evaluated their performance on the breast histopathology images dataset as CNNs, VGG16, and knowledge-based methods used with three different classifiers, including Neural Network, Multilayer perceptron, and Random Forest. Their work suggests that the knowledge-based CAD system achieved the best performance among the presented approaches. According to my opinion, the major revision is needed before acceptance for publication in Cancers.
Major comments:
1. Why do the authors focus on breast pathology images for this study? Compared to other pathologies, does breast pathology possess any unique value that warrants the exclusive training of an automated diagnostic model? If breast pathology is not particularly distinctive, did the authors merely employ methodological transfer due to the availability of a breast pathology dataset?
2. Why do the authors choose VGG16 among many deep learning models? There are many better models than VGG16, for example, Resnet50 and VIT.
3. What are the advantages of the model considered in the paper compared with DsHoNet and EFML?
4. In line 190, "The obtained features were fed to three classifiers: a neural network, Multilayer Perceptron (MLP), and Random Forest (RF)." Why were only these three classifiers—neural network, Multilayer Perceptron, and Random Forest—chosen for performance comparison? Evidently, numerous classifiers in machine learning conform to your data distribution. If there is no rationale for selecting solely these three classifiers, it could be perceived as result-driven bias.
5. Beginning with line 232, in the "C) Knowledge-based feature extraction:" section, is the calculation of three sets of features scientifically grounded and justified? Why not directly employ the original 33 features for model training?
6. Does the proposed method exhibit satisfactory performance solely on the BreakHis 400X Histopathology image dataset? Given that disease subtypes exhibit a high degree of specificity and clustering effects due to regional variations, please validate the predictive capabilities of the model using an external validation dataset.
Minor comments:
7. The figures and captions in this article display a significant degree of arbitrariness. Figures should be supplemented with complete captions, such as "Figure 2. Biomarkers of benign and malignant histopathology images." In images (e) and (f), conspicuous differences in markings are present, but the meanings of these variations remain unspecified. The same biomarkers in images (e) and (f) should be compared at identical positions, and explanations and interpretations of the selected biomarkers should be furnished. Furthermore, the numbering of Figure 2's captions should not commence with (e) and (f); instead, it ought to begin anew with (a) and (b).
8. The drawing method of Figure3 is too messy and text is incomplete, especially VGG16 Architecture. Figure5 has the same problem.
9. Different methods are not analyzed in the experiment, and the highlights of the method in this paper are not explained.
10. Too much words are used to describe informative work, and there are some repeated description of applications of deep learning and machine learning.
11. The module that is suggested to be deleted should not appear in the model structure diagram.
12. The structural ablation experiment is not perfect.
13. Is the method of this paper three CADs? Or are there three branch structures for comprehensive analysis? The expression is not clear, and Figure3 is easy to mislead readers.
14. How do you ascertain that the deep learning method employed has reached optimal convergence during the training process? Please elaborate on the specific results.
15. The feature extraction method based on knowledge-based feature sets has been demonstrated to have superior training performance on this dataset compared to other extraction methods. Is this result constrained by the size of the pathology dataset in question?
16. Please elucidate in the Conclusion in a clear and detailed manner the advantages and innovative aspects of your work.
Author Response
We thank you for the opportunity to revise “Deep Learning- and Expert Knowledge-based Feature Extraction and Performance Evaluation in Breast Histopathology Images”. We are pleased to submit the revised manuscript. We appreciated the constructive criticisms of the reviewers. This document outlines how we have addressed them. All the revisions are written in blue in the manuscript.
---------------------------------------------------------------------------------------------------------------------------------------
Reviewer 2:
Major comments:
- Why do the authors focus on breast pathology images for this study? Compared to other pathologies, does breast pathology possess any unique value that warrants the exclusive training of an automated diagnostic model? If breast pathology is not particularly distinctive, did the authors merely employ methodological transfer due to the availability of a breast pathology dataset?
Thank you. We conducted this work as a continuation of reference 9, which proposed knowledge-based feature extraction to detect benign and malignant tissues. We also validated the features using the BreakHis dataset, which is public domain and widely used by other researchers. It is the reason we continued using the BreakHis dataset. The dataset is also the most commonly employed in breast pathology classification studies. The proposed methodologies can be applied to other pathology images too. Indeed, we tested our methods on the Acute Lymphoblastic Leukemia (ALL) image dataset, a public-domain dataset with three classes. We acquired a similar performance to the BreakHis dataset. However, we did not include the Acute Lymphoblastic Leukemia (ALL) dataset results in our work to solely focus on breast cancer detection since the authors work in breast cancer research.
- Why do the authors choose VGG16 among many deep learning models? There are many better models than VGG16, for example, Resnet50 and VIT.
We appreciate your comment regarding our choice of using VGG16 other than the various deep-learning models available. We understand your viewpoint that other models, such as ResNet50 and VIT, may be considered superior options. While it is true that ResNet50 and VIT are indeed better models, VGG16 has its own merits, making it a suitable choice for our research. VGG16 is a deep convolutional neural network, and its simplicity and straightforward design make it easier to understand and implement, which was beneficial for our research objectives. Additionally, VGG16 has demonstrated strong performance across various image classification benchmarks, and its pre-trained weights are readily available, allowing for efficient transfer learning.
While ResNet50 and VIT are undoubtedly advanced models that have achieved remarkable results in many applications, they may not always be the best fit for every scenario. The choice of deep learning model often depends on the dataset's specific characteristics, the problem's complexity, and the available computational resources. Although many studies proved that the Resnet50 model performs better than VGG16, detecting errors becomes difficult due to its deeper network. Furthermore, if the network is shallow, the learning might be inefficient. Medical image datasets need to build efficient learning models to predict the classes. In our research, we wanted to use VGG 16 due to its very small receptive fields instead of massive fields, like Resnet50, for the BreakHis dataset.
We are implementing the Vision transformer (VIT) method for breast cancer diagnosis from the histopathology images as an ongoing work.
We added a paragraph in Section 6, discussing the choice of the VGG-16 algorithm in our work.
- What are the advantages of the model considered in the paper compared with DsHoNet and EFML?
We have added the following discussion in Section 6.
Our work's models have less complexity than DsHoNet and EFML. The research approach of using a dual-stream high-order network (DsHoNet) for breast cancer classification can be considered efficient in terms of its potential to improve accuracy and performance. By employing a dual-stream architecture, the model can leverage multiple sources of information or features extracted from histopathology images, enhancing the classification process. Additionally, incorporating high-order network components allows for capturing complex relationships and patterns within the data, potentially leading to improved classification results. However, the actual efficiency of this research would depend on factors such as the size and diversity of the dataset, the training methodology, and the evaluation metrics used to assess the performance of the classification model.
However, DsHoNet can generally be considered a relatively complex model compared to more straightforward approaches. The use of a dual-stream architecture implies the presence of two separate pathways or streams within the network, each handling different types of input or extracting different kinds of features. It introduces additional complexity compared to single-stream models.
Furthermore, incorporating high-order network components suggests the model captures higher-order interactions or dependencies among features. It typically involves modeling complex relationships and may require more computational resources and advanced techniques.
Overall, DsHoNet is likely to have a higher complexity level than traditional models, as it aims to leverage multiple streams of information and capture more intricate relationships within the data.
EFML involves designing a network architecture that integrates and learns from multiple modalities or feature sources. It typically requires the development of specialized network modules or fusion strategies to combine information from different sources. The complexity of the architecture depends on factors such as the number and type of modalities, the fusion approach used (early fusion, late fusion, or intermediate fusion), and the interactions between modalities within the network.
EFML involves training a model with multiple modalities or feature sources, which increases the computational requirements compared to single-modal approaches. The training process may include backpropagation and optimization across multiple branches or streams, potentially leading to increased training time and memory usage. Additionally, the computational complexity can increase if the modalities have different resolutions or require separate preprocessing steps.
- In line 190, "The obtained features were fed to three classifiers: a neural network, Multilayer Perceptron (MLP), and Random Forest (RF)." Why were only these three classifiers—neural network, Multilayer Perceptron, and Random Forest—chosen for performance comparison? Evidently, numerous classifiers in machine learning conform to your data distribution. If there is no rationale for selecting solely these three classifiers, it could be perceived as result-driven bias.
Thank you. We appreciate your concerns about potential bias and understand the importance of a comprehensive analysis. We intend to present an unbiased and thorough evaluation of the classifiers in the given context.
We selected the classifiers based on their popularity among researchers. The neural network (NN), Multilayer Perceptron (MLP), and Random Forest were chosen due to their popularity and effectiveness in handling various data types. These classifiers represent different approaches to classification and are known for their robust performance. By comparing these classifiers, we aimed to provide insights into the strengths and weaknesses of other classification methods.
We added KNN, SVM, NNN, and Decision tree classifiers to our existing classifiers to address your concern and suggestion. The associated additions are made throughout the paper.
- Beginning with line 232, in the "C) Knowledge-based feature extraction:" section, is the calculation of three sets of features scientifically grounded and justified? Why not directly employ the original 33 features for model training?
Thank you. The original 33 features were validated by [9]. We directly fed the features to seven classifiers for training. It is explained in Section 4C. We revised the methodology section (page 8) to clarify it further.
- Does the proposed method exhibit satisfactory performance solely on the BreakHis 400X Histopathology image dataset? Given that disease subtypes exhibit a high degree of specificity and clustering effects due to regional variations, please validate the predictive capabilities of the model using an external validation dataset.
Thank you. We intentionally used only the BreakHis 400x dataset for a couple of reasons. 1) Our work compares the knowledge-based features with CNN and VGG-16-based features. Knowledge-based features are defined for histopathology images with 400x magnification since it shows the nuclei characteristics and segmentation of nuclei from 400x is still challenging due to overlapping and clustered nuclei. 2) The dataset provides binary classification, and our work proposed three models for binary classification.
We are aware that there are other datasets, such as the BACH dataset, which has four classes, including normal, benign, in-situ, and invasive. It is a small dataset with 430 images with no defined magnification value. Our research work was designed for a binary classification as benign or malignant. To be able to use the BACH dataset, the proposed CAD architecture needs to be redesigned. Even if it is done, the discussion section will be challenging to compare binary and multi-class results.
For these reasons, we limited our work to the BreakHis dataset. Table 2 lists the previous studies on binary classification using the 400x dataset. We believe it provides an objective comparison platform. We added a paragraph in Section 6 explaining our reasons for using the BreakHis 400x dataset.
Minor comments:
- The figures and captions in this article display a significant degree of arbitrariness. Figures should be supplemented with complete captions, such as "Figure 2. Biomarkers of benign and malignant histopathology images." In images (e) and (f), conspicuous differences in markings are present, but the meanings of these variations remain unspecified. The same biomarkers in images (e) and (f) should be compared at identical positions, and explanations and interpretations of the selected biomarkers should be furnished. Furthermore, the numbering of Figure 2's captions should not commence with (e) and (f); instead, it ought to begin anew with (a) and (b).
Thank you. We corrected the Figures.
- The drawing method of Figure 3 is too messy, and the text is incomplete, especially VGG16 Architecture. Figure 5 has the same problem.
Thank you. We revised the figures.
- Different methods are not analyzed in the experiment, and the highlights of the method in this paper are not explained.
Thank you. We expanded the experimental results section and discussed the works in detail.
- Too much words are used to describe informative work, and there are some repeated descriptions of applications of deep learning and machine learning.
Thank you. We edited the paper to remove repetitive sections. Hopefully, those repeated descriptions were minimized in the revised manuscript. We will gladly revise any repetitive sections if you expressly point them out.
- The module that is suggested to be deleted should not appear in the model structure diagram.
Thank you. We removed the mentioned section.
- The structural ablation experiment is not perfect.
Thank you. We hope the revised manuscript will improve the presentation and explanation of the experiment. We will gladly clarify any concerns about the experimental work if the reviewer provides a specific concern.
- Is the method of this paper three CADs? Or are there three branch structures for comprehensive analysis? The expression is not clear, and Figure3 is easy mislead readers.
Thank you. Yes, the paper designed three CADs. To avoid confusion, we revised Figure 3 by separating the CADs and presenting them in the associated section.
- How do you ascertain that the deep learning method employed has reached optimal convergence during the training process? Please elaborate on the specific results.
Thank you. We added figures showing the convergence rates for the training and validation. The additions can be found in Section 6.
- The feature extraction method based on knowledge-based feature sets has been demonstrated to have superior training performance on this dataset compared to other extraction methods. Is this result constrained by the size of the pathology dataset in question?
No, the size of the dataset does not affect the feature calculations. The features' mathematical model was developed based on content knowledge. The classifier's performance may change because of the type of classifier design and specifications, such as support vector machines, KNN, ANN, CNN, etc. However, we can employ the knowledge-based features to design a rule-based classifier such as a Fuzzy classifier. So the dataset size does not affect the feature calculations.
On the other hand, dataset size affects the features calculated via CNN and VGG-16. It is the major disadvantage of these two models. To highlight the limitation of such models, our study compares the capabilities of the three feature sets coming from traditional CNN, transfer learning, and knowledge-based.
We added a paragraph in Section 6 to elaborate on it.
- Please elucidate in the Conclusion in a clear and detailed manner the advantages and innovative aspects of your work.
Thank you. We edited the Conclusion section.
Reviewer 3 Report
Dear Authors
The development of breast cancer diagnosis methodology is an extremely important problem. In this paper used three approaches to extract the features from the images. The Authors undertook this task and presented the results in an interesting article entitled: Deep Learning- and Expert Knowledge-based Feature Extraction and Performance Evaluation in Breast Histopathology Images.
This interesting paper evaluates three CAD systems and their performance in breast cancer diagnosis: a Convolutional Neural Network, transfer learning architecture and knowledge-based features based on domain expertise using Neural Network, Random Forest, and Multilayer perceptron classifiers. The convolutional layer has digital filters performing the convolution operation on the images. The results of this study showed that the 440 knowledge-based feature set outperformed the CNN and VGG16-based systems. It is 441 likely because the high-quality features of a histopathology image dataset were extracted 442 from each cell nuclei to distinguish benign and malignant cases.
Research findings supporting the diagnosis of malignant and benign forms of cancer certainly deserve publication.
Yours faithfully, Reviewer.
Author Response
Thank you for your review.
We thank you for the opportunity to revise “Deep Learning- and Expert Knowledge-based Feature Extraction and Performance Evaluation in Breast Histopathology Images”. We are pleased to submit the revised manuscript. We appreciated the constructive criticisms of the reviewers. This document outlines how we have addressed them. All the revisions are written in blue in the manuscript.
Round 2
Reviewer 1 Report
Line 35: Instead of: “…excluding nonmelanoma…” replace with “….excluding nonmelanoma skin cancer….”
Line 60: Apoptosis is not cell death, is programmed cell death. There are other types of cell death.
Line 91: What SIFT, SURF and ORB mean?
Line 226: “…the details were presented in the next section.” The verb doesn´t match the meaning of the sentence.
Line 498: The values in Table 1 don´t appear to be the same as in Table 2.
Regarding Q6:
The authors can keep their work focusing in a binary classification, renaming the normal and benign of the BACH dataset as benign, and invasive as malignant. The magnification was 200x.
Regarding Q7/Q9: Line 348 “Details of the features can be found in [9].” I still think there should be a description of the features. I still didn´t learn in this paper what were the features that more significantly helped in the distinction between benign and malignant cells. At least feature 27, 31, 18, 24 and 30, the 5 most relevant.
Minor editing of English language required
Author Response
5/31/2023
Title: Deep Learning- and Expert Knowledge-based Feature Extraction and Performance Evaluation in Breast Histopathology Images
REVISION NOTES FROM AUTHORS
We thank you for the opportunity to revise "Deep Learning- and Expert Knowledge-based Feature Extraction and Performance Evaluation in Breast Histopathology Images." We are pleased to submit the revised manuscript. We appreciated the constructive criticisms of the reviewers. This document outlines how we have addressed them. All the revisions are done using the manuscript's "track changes" function.
---------------------------------------------------------------------------------------------------------------------------------------
Reviewer 1:
Comments:
- Line 35: Instead of: “…excluding nonmelanoma…” replace with “….excluding nonmelanoma skin cancer….”
Thank you for your comment. We changed the sentence according to your suggestion.
- Line 60: Apoptosis is not cell death, is programmed cell death. There are other types of cell death.
Thank you. We changed the "cell death" to "programmed cell death".
- Line 91: What SIFT, SURF and ORB mean?
Thank you for your comment. We revised the sentence.
- Line 226: "…the details were presented in the next section." The verb doesn't match the meaning of the sentence.
Thank you for your suggestion. We revised the sentence.
- Line 498: The values in Table 1 don't appear to be the same as in Table 2.
Thank you. We updated the values in Table 2.
We want to explain the reason behind the slightly different numbers. Supervised classifiers have a training stage that selects data randomly. Every time we run it, the performance may have an insignificant change in performance metrics. While we worked on revising our paper, we reran the algorithm. The numbers in the table represented the decimals rounded to the nearest hundredth. The following example explains it.
Recall=0.824 rounded to 0.82
Recall=0.826 rounded to 0.83
- Regarding Q6: The authors can keep their work focusing in a binary classification, renaming the normal and benign of the BACH dataset as benign, and invasive as malignant. The magnification was 200x.
Thank you. We will consider your suggestion using the BACH dataset to evaluate our method after we modify our method for multiclass classification. We are unwilling to use the BACH dataset by labeling the normal and benign classes as benign. Such action will prevent us from comparing the results with other works in the literature. We hope you find our work satisfactory in its current state.
- Regarding Q7/Q9: Line 348 "Details of the features can be found in [9]." I still think there should be a description of the features. I still didn't learn in this paper what were the features that more significantly helped in the distinction between benign and malignant cells. At least feature 27, 31, 18, 24 and 30, the 5 most relevant.
Thank you. Section 4C, pages 8 through 11, presents the mathematical equations and description of the knowledge-based features. We think the following sentence, "Details of the features can be found in [9]." in the 4C gave the impression that the descriptions were insufficient. We directed the readers to the reference [9] as graphical illustrations show the feature characterizations. Because of copyright issues, we are not able to include those figures in our manuscript. The reference [9] is published in IEEE CBMS conference proceedings and can be accessed online.
Reviewer 2 Report
The authors have done well in revising their manuscript. My recommendation therefore is acceptance in present form.
Author Response
5/31/2023
Title: Deep Learning- and Expert Knowledge-based Feature Extraction and Performance Evaluation in Breast Histopathology Images
REVISION NOTES FROM AUTHORS
We thank you for the opportunity to revise “Deep Learning- and Expert Knowledge-based Feature Extraction and Performance Evaluation in Breast Histopathology Images”. We are pleased to submit the revised manuscript. We appreciated the constructive criticisms of the reviewers. This document outlines how we have addressed them. All the revisions are done by using “track changes” function in the manuscript.
---------------------------------------------------------------------------------------------------------------------------------------
Reviewer 2:
The authors have done well in revising their manuscript. My recommendation therefore is acceptance in the present form.
Thank you for your constructive comments.